# Characterization and Antimicrobial Activity of *Nigella sativa* Extracts Encapsulated in Hydroxyapatite Sodium Silicate Glass Composite

**DOI:** 10.3390/antibiotics11020170

**Published:** 2022-01-28

**Authors:** Salima Tiji, Mohammed Lakrat, Yahya Rokni, El Miloud Mejdoubi, Christophe Hano, Mohamed Addi, Abdeslam Asehraou, Mostafa Mimouni

**Affiliations:** 1Applied Chemistry and Environment Laboratory, Faculty of Sciences, Mohammed First University, Oujda 60000, Morocco; m.mimouni@ump.ac.ma; 2Solid Mineral Chemistry Laboratory, Faculty of Sciences, Mohammed First University, Oujda 60000, Morocco; mohammed.lakrat@um6p.ma (M.L.); e.majdoubi@ump.ac.ma (E.M.M.); 3High Institute of Biological and Paramedical Sciences, ISSB-P, Mohammed VI Polytechnic University (UM6P), Benguerir 43150, Morocco; 4Bio-Resources, Biotechnology, Ethno-Pharmacology and Health Laboratory, Faculty of Sciences, Mohammed First University, Oujda 60000, Morocco; y.rokni@usms.ma (Y.R.); asehraou@yahoo.fr (A.A.); 5Research Unit Bioprocess and Biointerfaces, Laboratory of Industrial Engineering and Surface Engineering, National School of Applied Sciences, Sultan Moulay Slimane University, 17 Mghila, Beni Mellal 23000, Morocco; 6Laboratoire de Biologie des Ligneux et des Grandes Cultures, INRA USC1328, Orleans University, CEDEX 2, 45067 Orléans, France; 7Laboratoire d’Amélioration des Productions Agricoles, Biotechnologie et Environnement (LAPABE), Faculté des Sciences, Université Mohammed Premier, Oujda 60000, Morocco; m.addi@ump.ac.ma

**Keywords:** *Nigella sativa* L., composite scaffold, hydroxyapatite, antimicrobial activity

## Abstract

*N. sativa* is an interesting source of bioactive compounds commonly used for various therapeutic purposes. Associate its seeds extracts with biomaterials to improve their antimicrobial properties are highly demanded. This study aims to investigate the encapsulation of *NS* extracts in hydroxyapatite nanoparticle sodium silicate glass (nHap/SSG) scaffold. *NS* essential oil (HS) was extracted by hydrodistillation, while hexane (FH) and acetone extracts (FA) were obtained using Soxhlet extraction. (FH) was the most abundant (34%) followed by (FA) (2.02%) and (HS) (1.2%). GC-MS chromatography showed that the (HS) contained beta cymene, alpha thujene, β-pinene and thymoquinone, while (FH) had mostly fatty acids and (FA) decane, 2.9-dimethyl, benzene 1,3,3-trimethylnonyl and beta cymene. Loaded nHap/SGG scaffolds with various amount of (FH), (HS) and (FA) at 1.5, 3, and 6 wt%; were elaborated then characterized by ATR-FTIR, X-ray and SEM techniques and their antimicrobial activity was studied. Samples loaded with 1.5 wt% HE was highly active against *C. albicans* (19 mm), and at 3 wt% on *M. luteus* (20 mm) and *S. aureus* (20 mm). Additionally, loaded scaffolds with 1.5 wt% AE had an important activity against *M. luteus* (18.9 mm) and *S. aureus* (19 mm), while the EO had low activities on all bacterial strains. The outcome of this finding indicated that loaded scaffolds demonstrated an important antimicrobial effect that make them promising materials for a wide range of medical applications.

## 1. Introduction

*Nigella sativa* L., also known as black cumin or black seed, is a plant that is bursting with interesting compounds responsible for medicinal proprieties [1]. Its seeds are rich of phytochemical components such as alkaloids, terpenes, flavonoids, polyphenols, and steroids known for their large spectrum of pharmacological potential [2,3]. The (FH) and (FA) of *N. sativa* seeds were previously tested for their wide array of therapeutics activities such anti-tumor [4], antimicrobial [3,5], anti-inflammatory [6], antioxidant [7], anticancer [8] and anti-diabetic activities [9]. In addition, pretreatment with *N. sativa* seeds have shown a protective effect against kidney injuries [10]. Furthermore, *N. sativa* was reported to be useful for treating infected bones and enhances the healing process as well as preventing for mitigating infection risk [11,12].

The hydroxyapatite nanoparticles (nHAp) are known by their structural and chemical similarity with biological apatite that represents around 70% of human bone mass [13]. This similarity justifies their natural biocompatibility, excellent bioactivity and adequate biodegradability [14,15]. Consequently, they are commonly applied in the form of powder, bulk, and coating, for various biomedical applications including bone tissue engineering, maxillofacial, dentistry and as coatings on metallic implants [16,17]. Importantly, nHAp-based materials are highly reactive and interact with biological entities forming chemical bonds with the adjacent biological tissue [18].

On the other hand, the lack of bactericidal properties of nHAp-based materials cannot prevent the attachment and growth of residual bacteria on the implant surface and develop biofilms, which can be responsible for implant-related infection [19]. In some cases, the bacteria cells involving orthopedic devices, may spread in the surrounding tissues and circulate in the whole body, which can potentially cause serious complications for patients with low systemic immunity [20]. Indeed, this implant-related infection can cause surgery revision and/or removal of the implant [21].

Thus, ideal biomaterials, when implanted, should induce osteogenic activity of osteoblasts while preventing infections and eradication of residual bacteria [22]. The fabrication of porous scaffolds with intrinsic antibacterial components is recognized as an effective strategy to treat traumatic bone injuries and prevent any contamination in bone defects [23].

The major drawback of nHap nanoparticles is their thermal instability. In fact, the conventional processes adopted for nHap consolidation are mainly based on high temperatures (>1000 °C), slow heating rates and long sintering duration, which is responsible for an extreme grain coarsening and high crystallinity with the formation of secondary phases [24] Therefore, this sintering technique destroys the structural stability of nHap as well as its biological activity, since the obtained phase is far from being similar to the natural bone [25].

Recently, a porous composite scaffold based on nHap and sodium silicate glass (SSG) was developed by a simple dehydration-drying process at near-room temperature. In this new process, sodium silicate solution was used as a mineral binder for the consolidation of nHAp without affecting their advantageous characteristics. The obtained results showed that the consolidated nHap/SSG scaffolds exhibited a structural and chemical composition close to the natural bone [25]. In addition, the in vitro biocompatibility confirms the non-toxicity of elaborated scaffold and can enhance attachment and proliferation of osteoblast-like cells that make it a promising candidate for bone healing applications. However, the potential application of this biomaterial as an antibiotic delivery system to prevent implant-related infection in bone defect sites has not been studied.

Thus, the combination of *N. sativa* essential oil and extracts with nHAp/SSG porous scaffolds stimulated the design of a new composite material with multiple characteristics, optimizing antimicrobial, and enhanced bioactivity.

The aim of this study is to explore the possibility to encapsulate the *N. sativa* essential oil and extracts, into nHAp/SGG composite and test the antimicrobial activity of loaded scaffolds for making it a suitable candidate as a biomaterial for bone healing applications.

## 2. Results

### 2.1. Extracts Yields

Yields were calculated based on starting extracted powder. Extract (FH) presented 34.56%, and extract (FA) had 2.03%. The (HS) of *N. sativa* presented a yield of 1.2%. Extract (FH) was the most abundant, followed by (FA) and (HS).

### 2.2. GC-MS Characterization

#### 2.2.1. The Essential Oil (HS)

The gas separation chromatography was carried out in 30 min. *N. sativa* essential oil had nine constituents with different proportionality. The most abundant compounds were β-cymene (38.05%) followed by α-thujene (13.70%) then thymoquinone (5.69%) (Table 1). The rest of compounds (α-pinene, β-pinene, ψ-cumene, β-cymene, γ-terpinene, aldehyde lilas, cyclohexen-1-ol, and carvacrol) represented less than 3%.

#### 2.2.2. Hexane Extract (FH)

Gas chromatography analysis of the extract (FH) gave eight compounds that presented different proportions in the global composition of the extract. Fatty acids were the most abundant components in extract (FH), with area peak value of 80.65% for linoleic acid, 2.96% for oleic acid, and 1.32% for palmitic acid (Table 2). E/Z nonadecatriene and ascorbic acid represented 6.24% and 4.39%, respectively.

#### 2.2.3. Acetone Extract (FA)

The volatile part of (FA) had eighteen constituents (Table 3). All peaks had different intensity, which means that the compounds have different proportions in the extract. The most abundant components were benzene 1,3,3-trimethylnonyl (21.62%), decane, 2.9-dimethyl (17.31%) and β-cymene (15.76%). Palmitic acid presented (7.29%) and alpha glyceryl linoleate had (6.85%). The other compounds presented less than 5%.

### 2.3. Scaffold Characterization

The XRD (X-ray diffraction) analysis is used to characterize the crystal structure of free and loaded nHAp/SGG scaffolds. The XRD pattern of free scaffolds (Figure 1) exhibited characteristic peaks of hydroxyapatite in the hexagonal crystal system (JCPDS No. 09–0432) [26]. Loaded scaffolds (Figure 1, HS, FA and FH) present the same free scaffold pattern without observing further secondary phases. This is probably due to the low quantity of encapsulated oils or to their amorphous nature. Moreover, the diffraction peaks broadening indicates the low crystallinity and nanoscale size of nHap particles in the developed scaffolds.

The (002) reflection peak from the different XRD patterns was used to determine the crystallite size of hydroxyapatite in the elaborated scaffolds from the Debye–Scherrer equation. The obtained results showed that the Hap particles have a mean value of 22 ± 3 nm, which confirms their nanometric range similar to the natural bone [27].

ATR-FTIR (attenuated total reflection-Fourier transform infrared spectroscopy) analyses were performed to further identify the chemical composition of the elaborated materials. The FTIR spectra of free and loaded scaffolds were given in (Figure 2). Regardless of the nature or the amount of plant extracts loaded, characteristic vibrational bands related to PO_4_^3−^ groups in the hydroxyapatite structure is observed in all composite scaffolds. The two bands detected at 560 cm^−1^ and 605 cm^−1^ (*υ*_4_) are corresponding to the symmetric bending mode of O-P-O [28]. The band observed near 960 (*υ*_1_), 1088 cm^−1^ (*υ*_3_) and 1023 cm^−1^ (*υ*_3_) are ascribed to the symmetric and asymmetric stretching modes of P-O in (PO_4_^3−^) groups, respectively [15]. The two bands located at the 864 cm^−1^ and 1460 cm^−1^ are attributed to carbonate ions (CO_3_^2−^), respectively [29]. The small and broad bands at 1650 cm^−1^ and 3480 cm^−1^ correspond to deformation and vibration of water molecules (H_2_O), respectively [30]. In addition, the weak band near 3560 cm^−1^ was assigned to stretching vibration of OH^−^ groups in hydroxyapatite structure [31].

Moreover, the FTIR spectrum of loaded composite scaffolds highlights the appearance of characteristic peaks of *N. sativa* oil between 2852 cm^−1^ and 2922 cm^−1^ ascribed to the symmetric stretching of C-H [32,33]. The two absorption bands at 1460 and 1376 cm^−1^ are attributed to symmetric orientation of -CH(CH_3_) and -CH(CH_2_), respectively [34]. The intensity of these absorption bands increased as the proportion of plant extracts increased.

In the case of *N. sativa* extracts, bands at 1128 cm^−1^ and 1084 cm^−1^ that correspond to -C-O elongation are usually observed, however it seems they are overlapping with PO_4_^3−^ groups in nHAp structure.

The SEM (scanning electron microscope) images of the free and loaded composite scaffolds are depicted in Figure 3. As it can be seen, all formulated scaffolds exhibit porous structures with pore size ranging from 50 to 200 µm. The cause of this porosity can mainly be attributed to the evaporation of water molecules during drying process [25].

It is important to mention that the formulated scaffolds are highly desired for bone tissue engineering, not only for their chemical and structural features similar to natural bone, but also for their micro and macroporosity [35]. In fact, the macroporosity is an essential characteristic for bone regeneration applications since it facilitates cell migration, diffusion of oxygen and nutrients for further bone mineralization [36].

### 2.4. Antimicrobial Activity Determination

The antimicrobial activity of loaded nHAp/SSG scaffolds was examined against the severe infection-causing pathogens, and the inhibition zone diameters are listed in Table 4. As it was expected, the nHAp/SGG scaffold DMSO, loaded and free of *N. sativa* extracts, used as negative control, shows no antimicrobial activity, as evidenced by the absence of inhibition zones, whereas the loaded scaffold exhibited an important inhibition zone.

Most bacteria strains are sensitive to 1.5 wt% encapsulation of (FH), and the highest inhibition diameter value in this case was found against *C. albicans* and *M. luteus* strains showing inhibition zone diameters of 19 ± 0.53 mm and 18 ± 0.46 mm, respectively. At 3 wt%, the measured inhibition zone diameter against *S. aureus* was 20 ± 0.62 mm. These results suggest that even at low concentration (1.5 and 3 wt%) of added plant extracts an important inhibition zone is developed, which indicates an efficient antimicrobial activity.

In the case of (FA), the best activities were noticed for *M. luteus* and *S. aureus* at 3 wt% (18.9–19 mm) like (FH). The encapsulated (HS) at 3 wt% activity was surprisingly high. In fact, for both strains of *M. luteus* and *S. aureus,* the activity was to the maximum (12.2–13 mm) but it was less pronounced to (FH) and (FA) extracts.

## 3. Discussion

*N. sativa* L. is a Mediterranean plant that presents a distinctive quantity of chemical compounds, depending on the geographical source of the plant [37] and the extraction methods [38]. Different yields of extract (FH) were reported from several countries as 37.33% [39], 36% [40], and 26% [41]. The (FA) had a yield of 2.5% [42], and (HS) presented a similar one of 1.2% [43]. However, some reported less important yields, ranging from 0.4% to 0.44% [44] and from 0.1% to 0.3% [45].

The components of *N. sativa* (HS) were similar to Hasanzadeh et al. [46]. The volatile oil composition was as carvacrol (2.2%), thymoquinone (2%), cymene (41.7%), longifolene (3.3%), and terpinol (1.9%). Several studies reported common components such as thujene, carvacrol, thymoquinone, and longifolene but they represented convergent results [43,44,47,48]. (FH) contained mostly fatty acids such as palmitic acid, linoleic acid, and oleic acid [39,49]. In fact, the plant is extremely rich of fat, so the matrix obtained from *N. sativa* seeds extraction by hexane is the highest [45].

The main compounds obtained in acetone extract were linoleic acid (53.60%), thymoquinone (11.80%), palmitic acid (10.53%), p-cymene (8.60%), longifolene (5.80%), carvacrol (3.70%), and 2.4 decadienal (1.40%). These compounds were majorly discovered in acetone extract of *N. sativa* [43]. However, compared to our results, thymoquinone, longifolene, and carvacrol were not found, and fatty acids were observably less abundant. These dissimilarities could be explained by the difference of extraction methods. In fact, our acetone extract was collected after several successive seed extractions using organic solvents with increasing polarity (hexane, chloroform, and ethyl acetate).

Due to the presence of this broad spectrum of natural bioactive agents, *N. sativa* extracts have been commonly applied for biological applications as anticancer, antidiabetic, antioxidant, antifungal, anti-inflammatory, and antibacterial agents. Furthermore, Ajita et al., report that *N. sativa* contribute effectively to stimulate the formation of new bone and enhance the socket healing process [50]. Additionally, *N. sativa* extract is used as an anti-osteoporosis agent and also to promote bone regeneration and healing due to the important amount of minerals such as calcium (Ca), phosphorous (P) zinc (Zn), strontium (Sr) and magnesium (Mg) [51,52].

Because of the presence of thymoquinone, p-cymene, and carvacrol, (HS) is a good antimicrobial [53,54]; in (FH), the antibacterial activity is related to oleic, linoleic, and palmitic acid derivatives [55,56], while in (FA), cysteine and ascorbic acid are good antimicrobials [57,58]. In order to benefit from the different biologically active components of *N. sativa*, the extracts of this herb were incorporated into the nHAp/SSG composite scaffold to make use of their important antibacterial characteristics, which will enhance the effectiveness of this composite material as a safe implant and avoid any bacterial infection after its implantation.

The loaded scaffolds are elaborated at near-room temperature through the dehydration-drying process, which enabled the consolidation of nHAp and the preservation of their characteristics close to the human bone, while encapsulating the plant extracts.

Antimicrobial efficacy of the loaded scaffolds was evaluated against targets, composed of one yeast strain (*C. albicans*), 3 g-positive bacteria (*M. luteus, S. aureus* and L. *innocua*) and 2 g-negative bacteria (*P. aeruginosa, E. coli*). The obtained results reveal that these scaffolds were able to inhibit the growth of bacteria spectrum and pathogenic yeast of *C. albicans*. The activity was greater against Gram-positive because this gram type is susceptible to the compounds of all extracts. Based on the quantity of extracts at (1.5%, 3%, and 6%) the inhibition of all bacterial strains always goes through a maximum then decreases. This could be explained by possible hydroxyapatite material saturation of the encapsulated extracts or because of a low diffusion of bioactive compounds into the pores of encapsulation materials, which can be sealed by high concentrations of the extract. In addition, all encapsulated extracts of *N. sativa* at 6% inhibit distinctively two bacterial strains *M. luteus* and *S. aureus*. We should notice that, based on our previous work [59], extract (FH) alone had an average activity, while (FA) alone was inactive for all tested bacterial strains. Surprisingly when encapsulated, their inhibitions were expressed in a strong way and the activity increased. This means that the encapsulation had a positive impact on the antibacterial activity. In fact, thymoquinone of *N. sativa* encapsulated in nanoparticles rose its anti-inflammatory and anti-proliferator activities [60]. Extract (FA), which did not give any antimicrobial effect before its encapsulation [59], in this study showed its activity expand against *C. albicans, M. luteus,* and *S. aureus*. The cause of that might be the process of encapsulation [60,61] and the near-room-temperature environment in which the biomaterial was prepared. However, (HS) activity diminished after its encapsulation [59]. This could be attributed to the exaggerated time of dry while the material was being prepared. In fact, the loaded materials consolidated at 37 °C for two weeks that could change the original chemical structures of the essential oil components.

Comparing the antimicrobial activity results of our loaded material with several studied biomaterials, the inhibition zone diameters of nHAp/SGG@FA 3 wt% against *E. coli* and *S. aureus* were higher or comparable to CS-TG/ZnO 8 wt% NC [62], CPCC + 10ORZ [63] and AAG [64] materials.

It can be concluded from the obtained results that the nHAp/SGG composite scaffold can be effectively loaded with a multitude of therapeutic biomolecules extracted from *N. sativa* and released without affecting, in all most cases, their biological properties especially their antimicrobial features. The loaded nHAp/SSG materials showed remarkable antimicrobial activity against pathogenic bacteria and yeast tested in this work which make them a promising candidate for bone healing applications.

## 4. Materials and Methods

### 4.1. Chemical Reagents

All of the organic solvents were of analytical grade (98.99%). Solvents, agar, di-ammonium hydrogen phosphate (NH_4_)_2_HPO_4_ and calcium nitrate tetrahydrate (Ca(NO_3_)_2_. 4H_2_O) were provided by Sigma Aldrich (Saint-Quentin Fallavier, France). Ammonium hydroxide solution (NH_4_OH 35%) was provided by Fisher Scientific (Illkirch, France).

The microbial strains were supplied by laboratory of Bio-resources, Biotechnology, Ethno-pharmacology and health at the University of Mohammed Premier (Oujda, Morocco).

### 4.2. Plant Material and Extracts

*N. sativa* seeds were furnished from a local market in Oujda, east of Morocco. An amount of 30 g of the plant was cleaned, reduced into powder and extracted by organic solvents (hexane, chloroform, ethyl acetate, and acetone) successively. The extractions were performed with the use of soxhlet apparatus for 24 h at 50 °C and the extracts were concentrated using rotavapor (BUCHI Rota vapor R-210, Büchi, Villebon-sur-Yvette, France) [65]. The essential oil was extracted by hydro distillation through hexane extract at 40 °C for 3 h.

### 4.3. GC-MS Analysis

The gas chromatography was produced for hexane extract (FH), volatile acetone extract (FA) and the essential oil from *N. sativa* (HS). The analysis was carried out by SHIMADZU instrument GCMS-QP2010 (Shimadzu, Noisiel, France) and computer controlled at 70 eV. The extracts were injected into a Rtx-5 Restek GC column (30 m × 0.25 mm, 0.25 μm) and the flow was 1.4 mL/min of helium gas. The temperature gradient program was from 50 to 200 °C at 10 °C/min and the scanning was for 30 min for (FH), (FA) and (HS). The ionization was maintained to 200 °C. The components characterization was conducted using mass fragments spectrums, and retention indices [66] of the compounds and the computer library NIST147 LIB [67].

### 4.4. Hydroxyapatite Nanoparticles Preparation

The hydroxyapatite nanoparticles were prepared using co-precipitation method as described in [25]. Briefly, an aqueous solution of di-ammonium hydrogen phosphate (0.24 M) is prepared and added drop wise under vigorous stirring into calcium nitrate tetrahydrate (0.35 M) solution. The pH was maintained around 10 during the reaction by adding ammonium hydroxide solution (0.5 M). Finally, the solution is left aging for 30 min and then filtered, washed with distilled water and dried at 80 °C overnight.

### 4.5. Loaded nHap/SSG Scaffolds Preparation

Loaded composite scaffolds were elaborated in three steps (Figure 4). In the first, homogenous solutions were prepared by the addition of extracted oils ((FH), (FA), and (HS)) form *N. sativa* into sodium silicate solution. The following three proportions of extracts were used in this investigation: 1.5, 3 and 6 wt%. The sodium silicate used in this study as a mineral binder had a molar ratio (SiO_2_/Na_2_O) of 1. In the second, nHap powder, was manually mixed with each solution until a homogenous malleable paste is obtained. The liquid-to-solid ratio used in this study was fixed at 0.5 cm^3^/g. Finally, obtained pastes were molded and oven-dried at 37 °C for 15 days. DMSO was loaded into nHap/SSG instead of extracts as negative control.

The loaded scaffolds were coded as nHAp/SSG@FH_x_, nHAp/SSG@FA_x_, nHAp/SSG@HS_x_ representing the extracted nature and its proportion.

### 4.6. Characterization of the Loaded Scaffolds

The crystallinity of precipitated n-HAp particles, and elaborated composite scaffolds were determined by X-ray diffraction (XRD) using Shimadzu XRD-6000 (Shimadzu, Noisiel, France) having a Cukα (λ = 0.154056 nm). The diffraction patterns were collected in the range of 2θ between 10° and 65° with a scanning speed of 4°/min.

The crystalline size of hydroxyapatite particles in the elaborated scaffolds was determined using the Debye–Scherrer formula (Equation (1)) from the respective xrd patterns. (ref: size effect of hydroxyapatite nanoparticles on proliferation and apoptosis of osteoblast-like cells.
(1)$$L=\frac{0.9\lambda }{\beta \;cos\theta }$$

L is the average crystallite size (nm), λ represents the X-ray wavelength (0.1544 nm), β is the full-width at half-maximum (FWHM), and θ represents the diffraction angle of the associated (hkl) plane.

Fourier Transform Infrared Spectroscopy in Attenuated Total Reflectance mode (ATR-FTIR) was carried out to identify the functional groups in the formulated composite scaffolds loaded by different extract *N. sativa*. The spectra were obtained in the range of 4000 to 470 cm^−1^ with a resolution of 4 cm^−1^, 256 scans, using FT/IR-4700 Spectrometer (FT/IR-4700, JASCO, Lisses, France).

Microstructure of loaded scaffolds was observed by scanning electron microscopy using a JEOL-JSM7001F apparatus (Croissy Sur Seine, France).

### 4.7. Evaluation of Antimicrobial Activity of Loaded Scaffolds

The antimicrobial activity of loaded nHap/SGG scaffolds were evaluated using agar disk diffusion method and the discs used where sterile Whatman paper discs. The strains used as targets were composed of yeast (*Candida albicans),* Gram-negative bacteria *(Pseudomonas aeruginosa, E. coli),* and Gram-positive bacteria *(Micrococcus luteus, Staphylococcus aureus,* and *Listeria innocua)*. Overnight cultures (10^7^ CFU/mL) of the strains obtained in Mueller Hinton (MH) broth was pour-plated on the surface of MH Agar, then the encapsulated materials, of 6 mm diameter at 1.5%, 3% and 6% (equivalent to 10, 20 and 40 μg/disc) were aseptically put on the agar culture medium. Dimethyl sulfoxide (DMSO) was used as negative control, and the plates were maintained at 4 °C during 2 h for pre-diffusion. After incubation of the cultures at 37 °C for 24 h for bacteria and at 30 °C for 24 h for yeast, the inhibition diameters obtained around disks were measured in millimeters [68]. All tests were repeated three times.

## 5. Conclusions

In this study, nHap/SSG composite scaffolds containing *N. sativa* extracts were formulated and their antimicrobial activity was investigated. ATR-FTIR analysis reveals that *N. sativa* extracts were successfully loaded into nHAp/SGG composite scaffolds. Additionally, the antimicrobial examination of showed that the elaborated materials could successfully inhibit the growth of a wide range of bacteria and the pathogenic yeast of *C. albicans*. Additionally, our findings have shown that the antimicrobial activity of composite scaffolds containing the hexane extract (FH), is more intense with higher inhibition diameter value compared to composite scaffolds containing acetone extract or essential oil (HS). The next step would be further tests of the loaded nHAp/SSG materials in vivo to validate the potential of these porous scaffolds for the clinical applications.

## Figures and Tables

**Figure 1 antibiotics-11-00170-f001:**
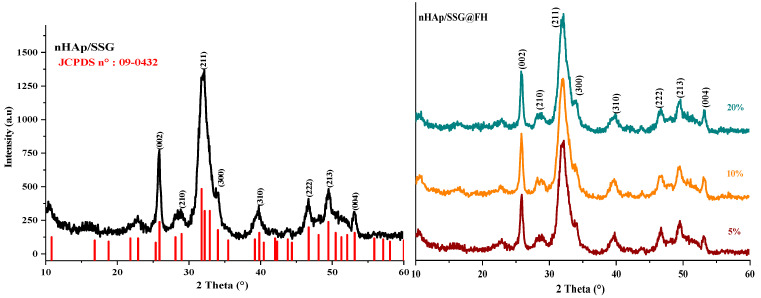
XRD spectrums of HAP with and without encapsulated extracts from *N. sativa*.

**Figure 2 antibiotics-11-00170-f002:**
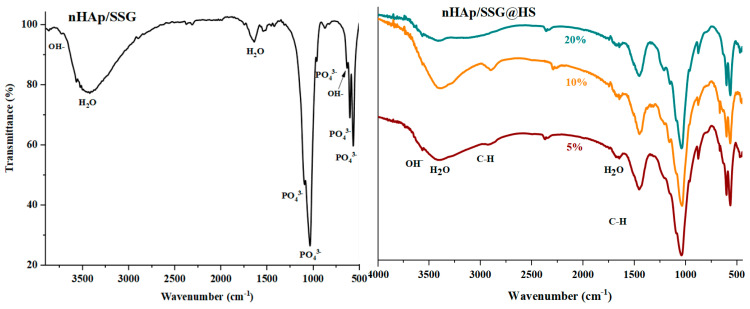
ATR-FTIR spectrums of free and loaded composite scaffolds.

**Figure 3 antibiotics-11-00170-f003:**
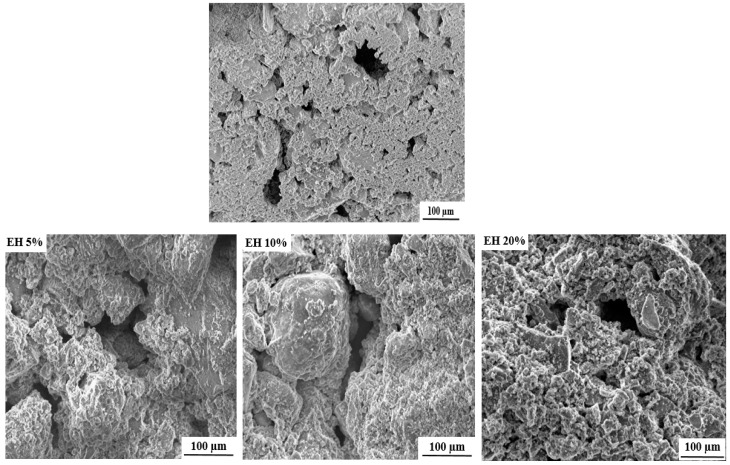
SEM images of free and loaded nHAp/SGG composite scaffolds.

**Figure 4 antibiotics-11-00170-f004:**
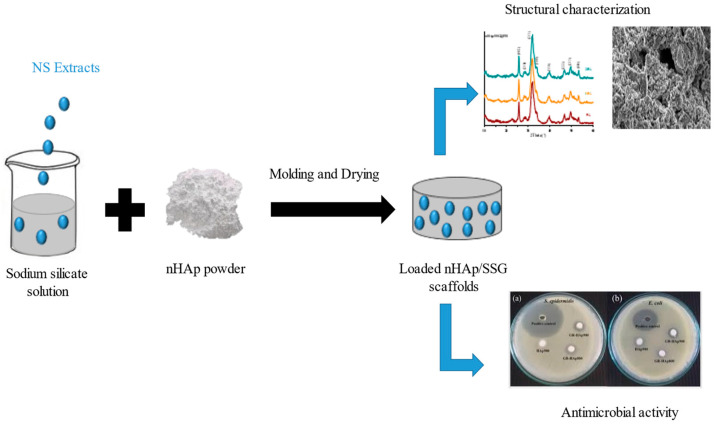
Loaded nHAp/SGG composite scaffold preparation.

**Table 1 antibiotics-11-00170-t001:** GC-MS Chemical composition of *N. sativa* L essential oil (HS).

Elution Order	Component	RT ^1^	% Area ^2^
1	Alpha-ThujeneOriganene (C_10_H_16_)	5.000	13.70
2	Alpha-Pinene (C_10_H_16_)	5.133	2.21
3	Beta-Pinene (C_10_H_16_)	5.842	2.19
4	1,2,4, trimethylbenzene, (C_9_H_12_).	6.100	1.30
5	Beta-Cymene (C_10_H_14_)	6.600	38.05
6	Gamma-Terpinene (C_10_H_16_)	7.158	0.69
7	Aldehyde lilac (C_10_H_16_O_2_)	7.792	0.55
8	Carvacrol (C_10_H_14_O)	8.175	2.19
9	Thymoquinone	10.233	5.69

^1^ RT: retention time; ^2^ % Area: percentage obtained by electronic integration measurement using a mass detector RT trace.

**Table 2 antibiotics-11-00170-t002:** GC-MS Chemical composition of *N. sativa* L. hexane extract (FH).

Elution Order	Component	RT ^1^	% Area ^2^
1	2.4-Decadienal	15.100	1.79
2	2-oxo-methyl ester Hexadecanoic acid	15.592	1.06
3	Phenol, 4-methoxy-2,3,6-trimethyl-	18.417	1.56
4	Palmitic acid, methyl ester	22.600	1.32
5	L(+)Ascorbic acid 2.6-dihexadecanoate	23.108	4.39
6	Oleic acid methyl ester	24.358	2.96
7	Linoleic acid	25.117	80.65
8	E/Z-1,3,12-Nonadecatriene	25.608	6.24

^1^ RT: retention time; ^2^ % Area: percentage obtained by electronic integration measurement using a mass detector RT trace.

**Table 3 antibiotics-11-00170-t003:** GC-MS Chemical composition of *N. sativa* acetone extract (FA).

Elution Order	Component	RT ^1^	% Area ^2^
1	Pentanoic acid, heptyl (C_12_H_24_O_2_)	4.47	2.72
2	1-Hepten-5-yne, 2-methyl-3-methylene (C_9_H_12_)	4.92	4.56
3	(R)-(2.2-dimethyl-1,3-dioxolane-4)methanol (C_6_H_12_O_3_)	5.14	3.28
4	Cumol (C_9_H_12_)	5.46	3.84
5	Psi-cumene (C_9_H_12_).	5.73	3.23
6	Benzene (1,3,3-trimethylnonyl) (C_18_H_30_)	5.95	21.62
7	beta.-Cymene (C_10_H_14_)	6.41	15.76
8	Decane, 2.9-dimethyl (C_12_H_26_)	7.50	17.31
9	1.3-Dioxolane-4-methanol,2,2-dimethyl,acetate (C_8_H_14_O_4_)	7.66	2.94
10	Dodecane (C_12_H_26_)	8.98	3.56
11	p-Cymen-3-ol (C_4_H_14_O)	10.53	1.84
12	Glycerine diacetate (C_7_H_12_O_5_)	11.13	1.88
13	Stearic acid (C_18_H_36_O_2_)	18.12	0.73
14	Palmitic acid (C_16_H_32_O_2_)	18.43	7.29
15	Linoleic acid (C_18_H_32_O_2_)	19.40	1.12
16	alpha.-Glyceryl linoleate (C_21_H_38_O_4_)	20.03	6.85
17	Oleic acid (C_18_H_34_O_2_)	20.07	0.56
18	Nonadecanoic acid (C_21_H_42_O_2_)	20.26	0.98

^1^ RT: retention time; ^2^ % Area: percentage obtained by electronic integration measurement using a mass detector RT trace.

**Table 4 antibiotics-11-00170-t004:** *N. sativa* (FH) (FA) and (HS) extract inhibition (in millimeters) at different percentages of encapsulation (1.5, 3 and 6 wt%) in nHAp/SSG materials against yeast, Gram-negative and Gram-positive strains.

Extracts	Inhibition Zones Diameter (in mm)									
		nHAp/SSG@								
Strain	Control	FH			FA			HS		
		1.5%	3%	6%	1.5%	3%	6%	1.5%	3%	6%
Yeast strain										
*C. albicans*	0 ± 0.23	19 ± 0.21	17 ± 0.25	12 ± 0.70	15 ± 0.56	13 ± 0.35	12 ± 0.42	11.5 ± 0.42	12 ± 0.70	11 ± 0.28
Gram negative										
*P. aeruginosa*	0 ± 0.28	11 ± 0.57	10 ± 0.21	07 ± 0.28	11 ± 0.85	10 ± 0.42	7.6 ± 0.28	11.7 ± 0.14	11 ± 0.57	8.9 ± 0.35
*E. coli*	0 ± 0.14	11 ± 0.35	11 ± 0.42	09 ± 0.14	11 ± 0.85	11 ± 0.85	9.5 ± 0.35	11 ± 0.14	12 ± 0.14	11 ± 0.28
Gram positive										
*M. luteus*	0 ± 0.45	18 ± 0.70	20 ± 0.85	15 ± 0.14	16 ± 0.42	18.9 ± 0.21	13.2 ± 0.70	11.9 ± 0.70	12.2 ± 0.21	11.3 ± 0.85
*S. aureus*	0 ± 0.31	12 ± 0.49	20 ± 0.35	11 ± 0.49	11 ± 0.28	19 ± 0.28	10 ± 0.70	12 ± 0.14	13 ± 0.07	11 ± 0.07
*L. innocua*	0 ± 0.65	12 ± 0.35	10 ± 0.85	09 ± 0.28	12 ± 0.28	11 ± 0.07	10.2 ± 0.78	12 ± 0.70	10.6 ± 0.14	9.3 ± 0.92

## Data Availability

All the data supporting the findings of this study are included in this article.

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
