# Peer review of "Characterization and Antimicrobial Activity of Nigella sativa Extracts Encapsulated in Hydroxyapatite Sodium Silicate Glass Composite"

_antibiotics, 2022, doi:10.3390/antibiotics11020170_

Round 1

Reviewer 1 Report

The manuscript by Salima Tiji et al., entitled as "Characterization and Antimicrobial Activity of Nigella sativa Extracts Encapsulated in Hydroxyapatite Sodium Silicate Glass Composite" is well written and executed. However, there are some minor concerns that need to be resolved before accepting. 

  • Make the uniformity in presentation of terminology used in manuscript like sometimes it has been mentioned as encapsulated materials (line 23 abstract), hydroxyapatite nanoparticle sodium silicate glass (nHap/SSG) composite scaffold (line 24 abstract),
  • Include the standard error or standard deviation in antimicrobial results.
  • Why scaffold term was used?
  • At Line 21-22, mentioned that “The aim of this study is to investigate the encapsulation of Nigella sativa extracts and evaluate the antimicrobial activity of the encapsulated materials”.
  • Put the word at Line 39-40, In summary Nigella sativa encapsulated materials…….
  • At line 94-97, it was mentioned that aim of the study as bone healing applications.
  • If the synthesized material is proposed as scaffolding material for bone healing applications (line 95-97), is there any related experiment carried out. The present study only describes the antimicrobial application which is different than bone healing related applications.

Material and method section

  • It is mentioned that antimicrobial activity was carried out through disc diffusion method. Which type of discs were used?
  • Include the final concentration of samples used for antimicrobial activity.
  • (Suppose if you are using 1% of sample solution and then 5 microliters were loaded on 6mm disc then concentration of loaded sample on disc will be 50 µg (50 µg/disc)
  • Is there only antimicrobial activity of loaded biomaterial? How about non loaded material, extract samples, and antibacterial reference drug?

Revise the table legends

Like in “table 4. it is written as Nigella sativa extracts activity at different percentages”. Is that the activity of different percentages of extracts? or extracts encapsulated materials   

Author Response

First of all, we would like to thank all the reviewers for their time and important remarks. we responded to questions, rectified errors and added missing parts.

Reviewer 1:

The manuscript by Salima Tiji et al., entitled as "Characterization and Antimicrobial Activity of Nigella sativa Extracts Encapsulated in Hydroxyapatite Sodium Silicate Glass Composite" is well written and executed. However, there are some minor concerns that need to be resolved before accepting. 

  • Make the uniformity in presentation of terminology used in manuscript like sometimes it has been mentioned as encapsulated materials (line 23 abstract), hydroxyapatite nanoparticle sodium silicate glass (nHap/SSG) composite scaffold (line 24 abstract),

We corrected this in the abstract

  • Include the standard error or standard deviation in antimicrobial results.

done

  • Why scaffold term was used?

We thank the reviewer for this question, in fact the term scaffold is used to mention the composite nature and to specify its porous structure that is very important for cell penetration, and adhesion and therefore promote new bone formation.

  • At Line 21-22, mentioned that “The aim of this study is to investigate the encapsulation of Nigella sativa extracts and evaluate the antimicrobial activity of the encapsulated materials”.
  • Put the word at Line 39-40, In summary Nigella sativa encapsulated materials…….

We added it

  • At line 94-97, it was mentioned that aim of the study as bone healing applications.

If the synthesized material is proposed as scaffolding material for bone healing applications (line 95-97), is there any related experiment carried out. The present study only describes the antimicrobial application which is different than bone healing related applications.

There was a study conducted to characterize the physicochemical properties, in vitro bioactivity, degradability and also cytotoxicity of the nHAp/SGG composite scaffold. The obtained results showed that the consolidated scaffolds exhibited a structural and chemical composition close to the natural bone as mentioned in reference.

[Lakrat et al. https://doi.org/10.1016/j.matchemphys.2020.124185.]

Furthermore, the composite scaffold presents an adequate porosity and mechanical profile required for bone healing applications. The in vitro cytotoxicity tests proved the non-hazardous and inductive nature of the fabricated composite over the MG-63 osteoblast-like cell line. In fact, the results show significantly enhanced cell proliferation.

In summary as reported by previous work mentioned above the nHAp/SGG composite scaffold could already be used in bone healing but here in this study we evaluate its antimicrobial activity when it encapsulates N. sativa so the material could heal the bones and inhibits microbial activity in defected sites at the same time, this can make our material suitable for bones healing applications.

Material and method section

  • It is mentioned that antimicrobial activity was carried out through disc diffusion method. Which type of discs were used?

The type of discs was used is Sterile Whatman paper Discs, we added at line 345

  • Include the final concentration of samples used for antimicrobial activity.
  • (Suppose if you are using 1% of sample solution and then 5 microliters were loaded on 6mm disc then concentration of loaded sample on disc will be 50 µg (50 µg/disc)

Those are final concentrations (10, 20 and 40 µg/disc) they are equivalent to 1.5%, 3% and 6% wt%. we added them as recommended: line 350

  • Is there only antimicrobial activity of loaded biomaterial? How about non loaded material, extract samples, and antibacterial reference drug?

We make the encapsulation into a solid and because there is no dissolution or recapitalization There is no unloaded quantity left so the totality of extract quantity is loaded into the material.

Revise the table legends

Like in “table 4. it is written as Nigella sativa extracts activity at different percentages”. Is that the activity of different percentages of extracts? or extracts encapsulated materials   

We have changed the legend and added details to understand the table 4: line 194-196

Reviewer 2 Report

The manuscript uses a good topic and has a good scientific structure. This manuscript is acceptable for publication in the journal after the following corrections:

Lines 48 & 51: Write the full scientific name of the plant only for the first time in full and in the next times write only the first letter of the genus in capital letters. Therefore, change "Nigella" to "N".

Line 90: Insert the reference number.

Line 91: change "Nigella" to "N".

Line 94: change "Nigella" to "N".

Line 123: change "table" to "Table".

Line 133: change "xrd" to "XRD".

Line 143: change "xrd" to "XRD".

LINE 203: The caption of Table 4 should be written completely and clearly. This caption is incomplete and does not reflect the results of the present study.

In addition, Table 4 does not show any statistical analysis and by placing lowercase and uppercase letters on the data, the effect of the studied factors (type and concentration of the loaded scaffolds) on the microorganisms should be shown.

This table is not scientifically acceptable.

Line 273-282: According to the results of Table 4, gram-negative bacteria were more sensitive to the loaded scaffolds than gram-positive bacteria. Why? Explain more.

Line 286: Rewrite the sentence "All organic solvents were grade np (98.99%)." This is not usual. For example, write: All of the organic solvents were of analytical grade … .

Line 294: Which parts of the plant did you use for this research? Did you mean to use the aerial parts of this plant? Did you use seeds too?

Line 295: change "grammes" to "grams".

Line 304: Mention the name and specifications of the column and the manufacturer?

Line 307-308: The amount of essential oil compounds in Table 1 is reported quantitatively and in μg/mL. How did you measure and report the quantitive of compounds? In this section, it is necessary to explain how to measure the quality and quantity of compounds. You can use the following references to write this section and add them to the list of sources:

Azhdarzadeh, Fatemeh, Mohammad Hojjati, and Saeed Tahmuzi Didehban. "Chemical composition and antimicrobial activity of Pelargonium roseum essential oil from southwest of Iran." Journal of Food and Bioprocess Engineering 1, no. 1 (2018): 33-38.

Zandi-Sohani, N., Hojjati, M., & Carbonell-Barrachina, A. A. (2012). Volatile composition of the essential oil of Callistemon Citrinus leaves from Iran. Journal of Essential Oil Bearing Plants15(5), 703-707.

Line 318: change "Nigella" to "N".

Line 352:  Where did you get the microbial strains?

Author Response

Reviewer 2 :

The manuscript uses a good topic and has a good scientific structure. This manuscript is acceptable for publication in the journal after the following corrections:

Lines 48 & 51: Write the full scientific name of the plant only for the first time in full and in the next times write only the first letter of the genus in capital letters. Therefore, change "Nigella" to "N".

done

Line 90: Insert the reference number.

It was the same study reported in Ref 25

Line 91: change "Nigella" to "N".

done

Line 94: change "Nigella" to "N".

done

Line 123: change "table" to "Table".

done

Line 133: change "xrd" to "XRD".

done

Line 143: change "xrd" to "XRD".

done

LINE 203: The caption of Table 4 should be written completely and clearly. This caption is incomplete and does not reflect the results of the present study.

We have changed to Table 4. Nigella sativa extracts activity at different percentages of encapsulation in nHAp/SSG materials against C. albicans, P. aeruginosa, M. luteus, E. coli, S. aureus and L. innocua.

In addition, Table 4 does not show any statistical analysis and by placing lowercase and uppercase letters on the data, the effect of the studied factors (type and concentration of the loaded scaffolds) on the microorganisms should be shown.

This table is not scientifically acceptable.

We rectified table 4 to be clear

Line 273-282: According to the results of Table 4, gram-negative bacteria were more sensitive to the loaded scaffolds than gram-positive bacteria. Why? Explain more.

Gram positive were more sensitive to loaded scaffolds we added and explained: line 247-248

Line 286: Rewrite the sentence "All organic solvents were grade np (98.99%)." This is not usual. For example, write: All of the organic solvents were of analytical grade … .

done

Line 294: Which parts of the plant did you use for this research? Did you mean to use the aerial parts of this plant? Did you use seeds too?

Those are the seeds it’s the most important part of the plant we montioned at the introduction line 38

Line 295: change "grammes" to "grams".

done

Line 304: Mention the name and specifications of the column and the manufacturer?

We added it line 296

Line 307-308: The amount of essential oil compounds in Table 1 is reported quantitatively and in μg/mL. How did you measure and report the quantitive of compounds? In this section, it is necessary to explain how to measure the quality and quantity of compounds.

We corrected it, table one reported %area not concentration the analysis was qualitative

Line 318: change "Nigella" to "N".

done

Line 352:  Where did you get the microbial strains?

It was mentioned on line 282-284

Reviewer 3 Report

"Characterization and Antimicrobial Activity of Nigella sativa Extracts encapsulated in Hydroxyapatite Sodium Silicate Glass Composite" present a biomaterial with an extract of black cumin with antimicrobial activity against pathogenic microorganisms. The approach to the experiment is interesting and has great potential and applicability. However, significant deficiencies have been detected when presenting and expressing the work results.

Authors are recommended to read the guide to publish papers in the journal to comply with the indicated format.

Specific comments/remarks:

-The bibliography of the work is poorly referenced. It begins with a numbering that is restarted on line 78 expressed in parentheses. I suggest to the authors that they revise the numbering and express it following the journal's rules. Similarly, many of the references are out of date.

-The abstract exceeds the recommended maximum of 200 words for the journal. The information provided will have to be reduced.-Line 74, use the nHap abbreviature.

-Line 79: I think you were trying to say: the obtained phase is far to be similar… instead of so fare.

-Line 102: If you define the abbreviation of hexane extract as FH, you don't need to refer continuously the abbreviation with parenthesis.

-Table 1: Here, you indicate the chemical composition of N. sativa essential oil. However, you stated a concentration, and in the material and methods section, you don't refer to any quantification in the GC analysis.

-Table 2 and Table 3: check that the names of the chemical compounds are written well. Make sure that the tables give the same information (for example, in table 3, you indicated LogP). As a suggestion, I will add the information about the identification of the compounds (LRIs calculated and those compared with the library).

-Line 131: I guess here it's 2.3 Scaffold characterization.

-Line 146: Add a reference supporting your results.

-Line 149: Figure 2

-Line 153: some units are missed

-Line 185: It is worth nothing to mention… Change to a more formal style. In the table, you don't specify what does (-) mean. Also, make sure that the information is not overlapped. As an example, you tell exactly the same in the footage and the table. Give the information of how your results were expressed.

-It is not easy to understand the discussion of the work. I suggest rewriting paragraph (lines 224-231) and paragraph (lines 251-259). Make sure that the references are actualized.

-It is interesting how the addition of the different extracts to the scaffolds had antifungal activity against some pathogens. However, I noticed that if you add a higher concentration of the extract, the antimicrobial activity decrease. Have the authors thought about an explanation of this?

-Line 293: Explain better how do you obtain the Nigella sativa extracts or reference a previous methodology.

-Line 307: How the LRIs indices were calculated?

-Line 336: Here you indicated equation 2. However, there's no equation 1 previously.

-Line 362: You explained a statistical analysis, but there's no statistical analysis in the figures and tables presented. Also, you indicated that you performed a one-way ANOVA, and a t-test, which is confusing. Explain better the section and include the statistical analysis in figures and tables.

Author Response

Reviewer 3:

"Characterization and Antimicrobial Activity of Nigella sativa Extracts encapsulated in Hydroxyapatite Sodium Silicate Glass Composite" present a biomaterial with an extract of black cumin with antimicrobial activity against pathogenic microorganisms. The approach to the experiment is interesting and has great potential and applicability. However, significant deficiencies have been detected when presenting and expressing the work results.

Authors are recommended to read the guide to publish papers in the journal to comply with the indicated format.

Specific comments/remarks:

-The bibliography of the work is poorly referenced. It begins with a numbering that is restarted on line 78 expressed in parentheses. I suggest to the authors that they revise the numbering and express it following the journal's rules. Similarly, many of the references are out of date.

Thank you for mentioning it We corrected those forms of citations

-The abstract exceeds the recommended maximum of 200 words for the journal. The information provided will have to be reduced.

We reduced the abstract to be suitable for the journal

-Line 74, use the nHap abbreviature.

done

-Line 79: I think you were trying to say: the obtained phase is far to be similar… instead of so fare.

corrected

-Line 102: If you define the abbreviation of hexane extract as FH, you don't need to refer continuously the abbreviation with parenthesis.

done

-Table 1: Here, you indicate the chemical composition of N. sativa essential oil. However, you stated a concentration, and in the material and methods section, you don't refer to any quantification in the GC analysis.

It was % area not a concentration We corrected it

-Table 2 and Table 3: check that the names of the chemical compounds are written well. Make sure that the tables give the same information (for example, in table 3, you indicated LogP). As a suggestion, I will add the information about the identification of the compounds (LRIs calculated and those compared with the library).

We corrected it and have delated logP to make all tables homogenic

-Line 131: I guess here it's 2.3 Scaffold characterization.

corrected

-Line 146: Add a reference supporting your results.

We added the reference

Bestimmung der Größe und der inneren Struktur von Kolloidteilchen mittels Röntgenstrahlen

-Line 149: Figure 2

Corrected

-Line 153: some units are missed

the missing units were added.

-Line 185: It is worth nothing to mention… Change to a more formal style. In the table, you don't specify what does (-) mean. Also, make sure that the information is not overlapped. As an example, you tell exactly the same in the footage and the table. Give the information of how your results were expressed.

We have changed the expression, (-): we delated it and replace it by a 0 to be clear, the footage was modified and how the results were expressed was added.

-It is not easy to understand the discussion of the work. I suggest rewriting paragraph (lines 224-231) and paragraph (lines 251-259). Make sure that the references are actualized.

We changed those two paragraph expressions.

-It is interesting how the addition of the different extracts to the scaffolds had antifungal activity against some pathogens. However, I noticed that if you add a higher concentration of the extract, the antimicrobial activity decrease. Have the authors thought about an explanation of this?

Lines 250-252: We tried to explain that by the saturation of scaffolds or by low diffusion of bioactive compounds into the pores of encapsulation materials, wish can be sealed by high concentrations of the extract.

-Line 293: Explain better how do you obtain the Nigella sativa extracts or reference a previous methodology.

We added a reference [40] of our previous work

-Line 307: How the LRIs indices were calculated?

the compounds were determined by comparing retention indices and spectral mass fragments given by the instrument (SHIMADZU instrument GCMS-QP2010) with computer library NIST147 LIB

and we added a reference on how those LRIs are calculated: [41]

-Line 336: Here you indicated equation 2. However, there's no equation 1 previously.

We corrected it to eq 1

-Line 362: You explained a statistical analysis, but there's no statistical analysis in the figures and tables presented. Also, you indicated that you performed a one-way ANOVA, and a t-test, which is confusing. Explain better the section and include the statistical analysis in figures and tables.

We chose to delete this part to be coherent

Round 2

Reviewer 3 Report

Thank you for correcting the errors and taking into account my suggestions.